# Seasonal Variability of the Acoustic Climate of Ski Resorts in the Aosta Valley Territory

**Christian Tibone [1],\*, Marco Masoero [2] , Filippo Berlier [1], Giovanni Tabozzi [2], Daniele Crea [1], Christian Tartin [1], Marco Cappio Borlino [1] and Giovanni Agnesod [1]**

[1] Aosta Valley Regional Environmental Protection Agency, loc. La Maladière 48, 11020 Saint Christophe, Italy; f.berlier@arpa.vda.it (F.B.); d.crea@arpa.vda.it (D.C.); c.tartin@arpa.vda.it (C.T.); m.cappioborlino@arpa.vda.it (M.C.B.); g.agnesod@arpa.vda.it (G.A.)

[2] Department of Energy, Politecnico di Torino, Corso Duca degli Abruzzi 24, 10129 Torino, Italy; marco.masoero@polito.it (M.M.); giovanni.tabozzi@googlemail.com (G.T.)

\* Correspondence: c.tibone@arpa.vda.it; Tel.: +039-0165-278573

**Abstract:** The Aosta Valley is an alpine region in north-west Italy that is characterized by a high level of naturalness, with extensive uninhabited areas that are distant from artificial sound sources. The Aosta Valley Regional Environmental Protection Agency (ARPA-VdA) has been particularly sensitive to the preservation of the soundscape, which is considered an integral part of the landscape, since the laws on noise pollution were first introduced. The nature of the ski areas in the Aosta mountains, which undergoes changes throughout the year, is surely of great importance, especially during the winter season, when the number of visitors is particularly high. In fact, during the winter, the sounds of nature are replaced by those produced by recreation and sports activities. Mountain and snow tourism, which are developed in sensitive environmental contexts in the Aosta Valley, are sectors of immense social and economic importance. Much of this tourism takes place in ski resorts. Three mountain areas with different characteristics, in terms of attendance and recreational/sport activities, have been examined in this paper, as part of a collaboration between ARPA-VdA and the Politecnico di Torino. Acoustic measurements were performed in order to identify the seasonal variations of sound emissions from both natural and anthropic sound sources. In addition to the standard environmental acoustic descriptors foreseen by European legislation ($L_{Aeq}$, $L_n$, $L_{den}$, etc.), the harmonica ($I_H$) index, which provides a quantitative evaluation of the acoustic quality on a zero to ten numerical scale, was used to qualify the acoustic climate of the three areas. The results presented in the paper provide useful information on a relevant subject—the preservation of the acoustic quality of a mountain area of touristic importance—which has been scarcely investigated so far.

**Keywords:** alpine region; seasonal tourist resorts; ski resorts; acoustic climate; soundscape; acoustic quality; harmonica index ($I_H$)

## 1. Introduction

Italian legislation [1] has always considered the protection of the most sensitive areas [2], including extensive open countryside areas, from the main sources of artificial noise [3] in its planning tools, as required by EU Directive 2002/49/EC on the management of environmental noise [4].

EU directive 2002/49/CE regulates the safeguarding of the natural soundscape [5]. This topic is of specific concerns in alpine regions, such as the Aosta Valley, which are characterized by, and appreciated for, their high values of environmental naturalness (Figure 1).

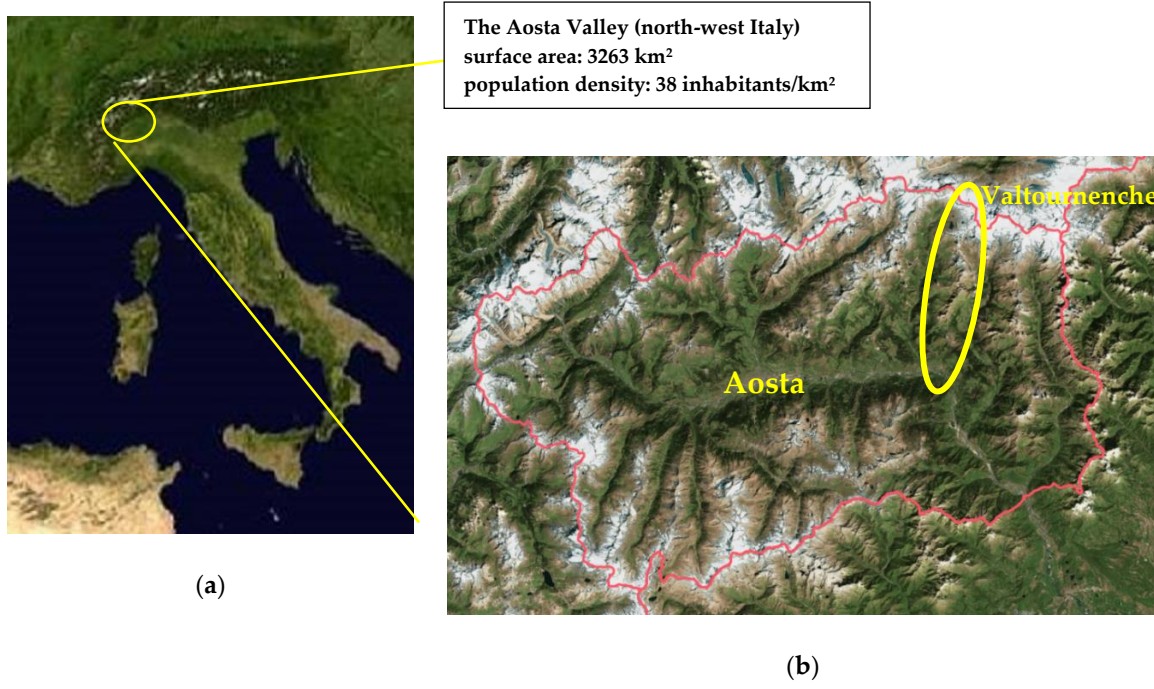

**Figure 1.** The Aosta Valley in north-west Italy (**a**), the smallest, least populous and least densely populated region in Italy; the yellow circle highlights the Valtournenche valley where the study was carried out (**b**) (Figure 1a courtesy of NASA/JPL-Caltech. Figure 1b is taken from the regional map website: http://geonavsct.partout.it/pub/GeoCartoSCT/index.html).

In this area, it is necessary to extend attention from built and highly infrastructured areas to remote ones, where the main feature is the natural quietness of the area, and the notion of sensitive noise receptor extends to the whole territory. The soundscape of mountain areas is determined by a great variety of sound sources. These sources include both natural sound emissions and sources linked to human activities, with natural environmental sound pressure levels ranging from the profound quietness of 25 dBA, in remote high-mountain snow fields in winter, to a high level of 70 dBA (or more) in the vicinity of water falls in summer. There is in fact a great variability of both natural and artificial sound levels in a natural environment, due to the variety of sources and to the time variability of the acoustic emissions, which is frequently of a seasonal type [6,7]. Hence, the impacts of artificial sources on the environmental sound pressure levels should be considered case by case, that is, by not referring to a fixed standard, but assuming the natural environmental sound level in each given site as the standard for the site itself.

Many ski resorts have a level of attendance that changes according to the time of year, as do the anthropic acoustic emissions [8]: for example, certain activities in mountain pastures require electrical current generators, chain saws, powered grass-mowers, etc. in summer, while the tourist and sports activities in winter involve cableways and ski lifts, snow cats, artificial snow making plants, etc. These artificial sources may cause annoyance and may sometimes overcome the natural environmental sound levels, thereby drastically modifying the natural acoustic climate.

At the same time, mountain and snow tourism are sectors of immense social and economic importance, which are developed in particularly sensitive environmental contexts. Much of this tourism takes place around ski resorts. The local mountain communities and administrations therefore have the difficult task of balancing the touristic valorization of the territory and preserving the acoustic quality of the natural areas. This task is made even more difficult by the limited availability of experimental data on the acoustic climate in mountainous areas of touristic interest.

This study, which was developed in collaboration between ARPA-VdA and the Politecnico di Torino, has two main goals:

- Firstly, to contribute to the knowledge on the characterization of the seasonal variations of the acoustic climate of three tourist resorts in Valtournenche, which is one of the main side valleys of the Aosta Valley. Valtournenche is crossed by the Marmore stream, which springs from the glaciers at the foot of one of the highest and most famous peaks in the Alps, the Matterhorn (named Cervino in Italian). The popularity of the valley is due to the presence of one of the most renowned ski areas in the western Alps (Cervino Ski Paradise), and to the possibility of practicing a variety of sports activities related to the mountain environment, such as climbing (the top of the Matterhorn represent one of the most popular international goals), trekking, hiking, and mountain biking in summer, ski mountaineering, and walking with snow rackets in winter.
- Secondly, to develop a method for the acoustic characterization of ski resorts through the identification of indicators that would be suitable to apply to the detected sound levels. The method could then be extended to other areas in the open countryside, as required by the European Noise Directive (END). Sound pressure levels were measured during the study to evaluate the seasonal variability of the acoustic climate in the selected areas. The acoustic quality of the areas was quantified through the harmonica index, $I_H$ [9], whose definition is provided in Section 2.7. The $I_H$ index, which is dimensionless and easy to interpret, even by non-technical operators, can be used to qualify the sound climate of an area over a scale ranging from zero to ten, taking into account the measured LAeq and LA95eq energy levels [10].

In addition to the collection of acoustic data in remote areas, the study has been aimed at providing additional criteria for the selection and the management of quiet areas in the open countryside, as required by the documents issued in recent years by the European Environment Agency [11].

## 2. Materials and Methods

### 2.1. The Study Sites

The Aosta Valley, the smallest region in Italy, with an area of 3263 km$^2$ and a population of about 130,000 inhabitants, has an almost completely mountainous territory with an average altitude above sea level of 2000 m, which also makes it the region with the lowest population density in Italy: 38 inhabitants/km$^2$. The main valley and thirteen other side valleys, which are the result of glaciations, represent the conformation of the entire regional territory. The distribution of the inhabitants is very irregular, and more than a third of the population is concentrated in the Aosta plain and in particular at the bottom of the valley, which is crossed by the Dora Baltea river and by the main transportation infrastructures: the railway, state roads 26 and 27, and the A5 Turin–Aosta motorway. The side valleys, the smallest of which have become considerably depopulated over the years, are characterized by a preponderance of rural and wooded areas, and of mountain pastures, as well as by the presence of vast high mountainous areas.

Three sites were chosen in Valtournenche (Figure 2), which, although close together, have different types of infrastructures and types of attendance [12]:

1.	Breuil-Cervinia is the most famous hamlet in Valtournenche; located at 2050m above sea level at the foot of the Matterhorn, it has 700 residents and offers a complete range of tourism activities, that is, heli-skiing, mountain biking, mountain climbing, and sonorous events. It is a very popular tourist resort in both winter and summer, with the presence of vehicular traffic, accommodation infrastructures and ski lifts starting from the center of the village and reaching the Plateau Rosà glacier at an altitude of 3500 m. The Breuil-Cervinia, Valtournenche and Zermatt area is one of the largest skiing areas in the Alps, with a varied "domaine skiable" which passes through three valleys in two countries, Italy and Switzerland, from the 3.883 m of the Piccolo Cervino down to the 1.524 m of the Valtournenche village.
2.	Chamois is the only car-free town in mainland Italy, and it can only be reached on foot, by bike or by cableway; it is an oasis of peace and quiet, and its environmentally friendly policy makes this

little alpine village overlooking the Matterhorn unique. The winter season in the Chamois ski area is a pleasant surprise for snow lovers, with more than 17 km of runs and its breath-taking view over the Matterhorn and snow-clad Alps.

3.    Cheneil, which is just a few kilometres away from Breuil-Cervinia and Valtournenche, is a small village in the Valtournenche municipality. The Cheneil basin is one of the few inhabited areas of the Aosta Valley that cannot be reached by car, but only on foot or by bike and, for a few years now, by a small and silent panoramic elevator that leads directly to the village. There are no ski lifts in the Cheneil basin. Ski mountaineering is practiced along the valley slopes in winter, and Cheneil is a destination for tourists who seek tranquility, or practice trekking and mountain climbing throughout the year. As a result of its remarkable territorial and urban integrity, Cheneil has maintained an unaltered and unique charm. It borders on the Chamois municipality, from which it can be reached on foot, by bike or on skis.

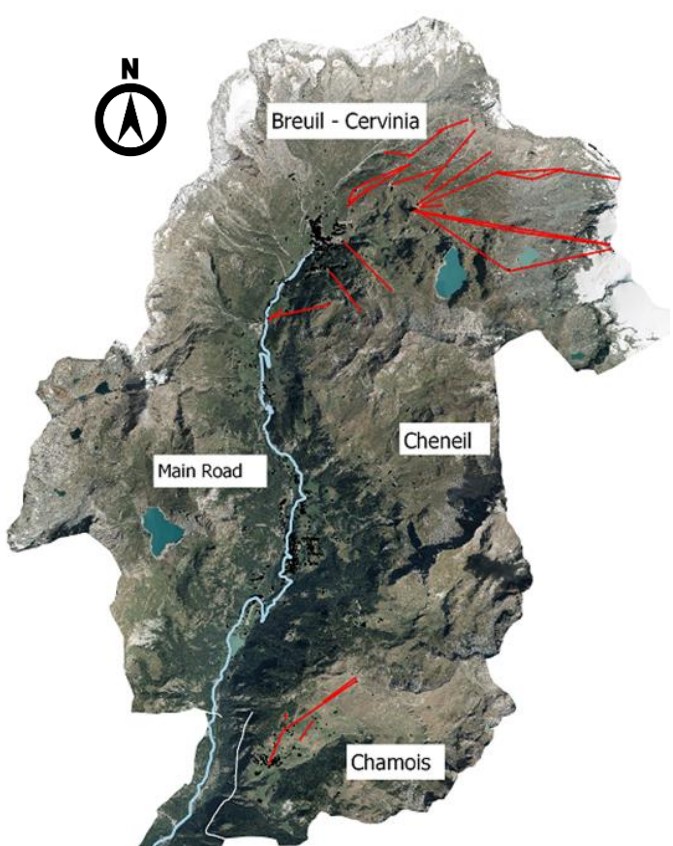

**Figure 2.** The tourist resorts in Valtournenche studied to evaluate the seasonal acoustic climate. The Breuil-Cervinia and Chamois ski lifts are in blue. Source: http://geonavsct.partout.it/pub/ GeoCartoSCT/index.html).

Table 1 reports the main characteristics of the tourist resorts studied in Valtournenche, in terms of the start/end of the expected 2019–2020 winter ski season, the minimum and maximum elevation above the sea level, the number of ski lifts and the length of the ski slopes [13].

**Table 1.** Main characteristics of the ski tourist resorts studied in Valtournenche.

| Name of the Ski Centre | Start/End of the 2019–2020 Winter Season | Altitude (m a.s.l.) Min. | Max. | Number of Ski Lifts | Length of the Ski Slopes (km) |
|---|---|---|---|---|---|
| Breuil-Cervinia, Valtournenche, Zermatt | from 26 October 2019 to 3 May 2020 | 1524. | 3883. | 53. | 350, of which 26.5 km is used for summer skiing |
| Chamois | 7–8 December 2019 and from 21 December 2019 to 29 March 2020 | 1815. | 2500. | 3 | 18 |
| Cheneil (Valtournenche) | / | 2105. | 3400. | / | 35 (only used for ski mountaineering) |

## 2.2. Measurement Areas

The acoustic measurement areas in the three Valtournenche tourist resorts were chosen on the basis of the main natural and artificial sound sources that are present, their accessibility and in order to cause the least interference possible with the ski-resort management activities (Figure 3).

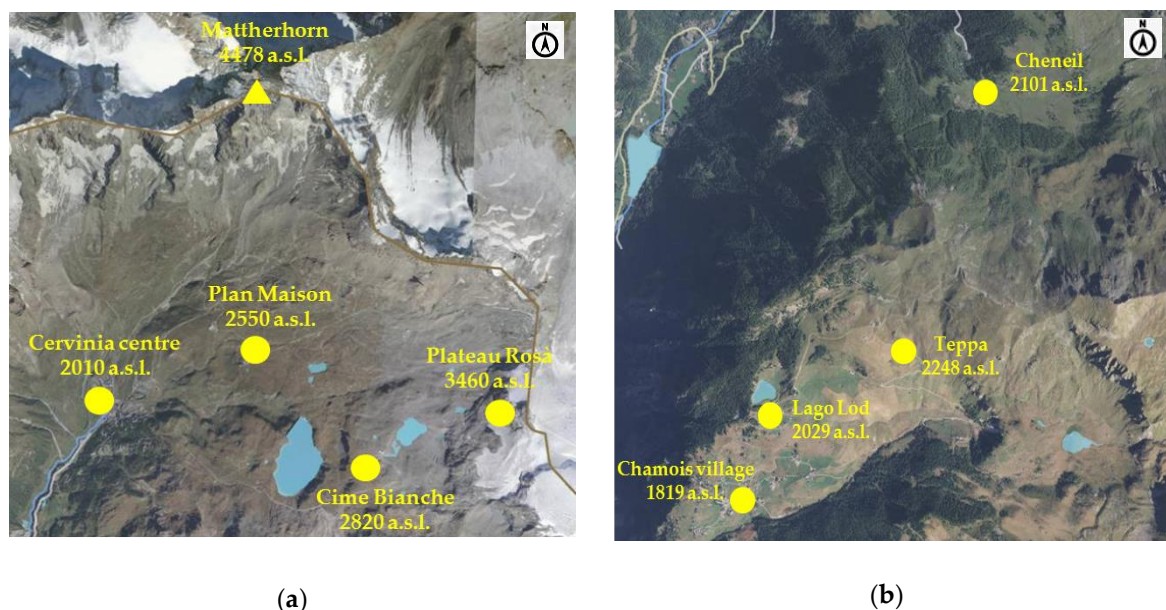

(**a**)                        (**b**)

**Figure 3.** Measurement areas chosen in (**a**) Breuil-Cervinia and (**b**) Chamois and Cheneil. Source: http://geonavsct.partout.it/pub/GeoCartoSCT/index.html.

The purpose of the study was to characterize and analyze the sound climate of the urban centers and to compare it with that of other remote sites within the relevant ski areas of the three resorts.

This was done for the Chamois and Breuil-Cervinia resorts, in which the areas have similar characteristics, but differ in terms of tourist attendance and altitude: the highest point in Chamois is located at about 2300 m, while in Breuil-Cervinia, it is at 3500 m, and is represented by the Plateau Rosà glacier. A single area, near some private houses, and an alpine refuge were chosen for the measurements in the Cheneil basin, due to the homogeneous landscape and territorial characteristics.

When no single sound source was found to prevail over the background level, the measurement points were sometimes chosen considering the general sound context.

## 2.3. Georeferencing and Characteristics of the Measurement Points

### 2.3.1. Chamois Village Area

Figure 4 shows the position and the name given to the acoustic measurement points on a map of Chamois. The main territorial characteristics around the measurement points are given in Table 2.

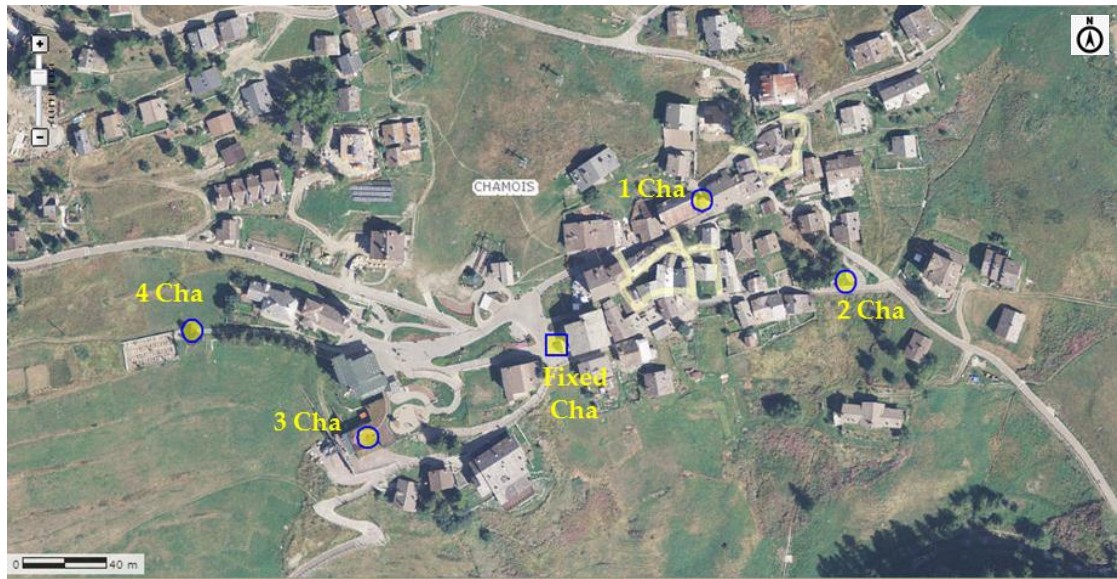

**Figure 4.** Chamois village, with the position and name given to the acoustic measurement points (square stands for the fixed monitoring point and circles for surrounding short-term monitoring points). Source: http://geonavsct.partout.it/pub/GeoCartoSCT/index.html.

**Table 2.** The Chamois village area: details of the locations of the acoustic measurement devices.

| Site | Measurement Point | Geographic Coordinates (WGS84 N/E) | | Altitude (m a.s.l.) | Feature of the Site |
|------|-------------------|------------------------------------|---|---------------------|---------------------|
| Chamois village | Fixed Cha | 45°50′17.16″ | 007°37′12.18″ | 1810 | Town hall balcony, main square |
| | 1 Cha | 45°50′19.50″ | 007°37′15.06″ | 1820 | Centre of the village, road going to Lod |
| | 2 Cha | 45°50′18.30″ | 007°37′18.36″ | 1815 | Rural road going to La Magdeleine |
| | 3 Cha | 45°50′07.74″ | 007°37′15.84″ | 1815 | Panoramic terrace, Cableway |
| | 4 Cha | 45°50′17.52″ | 007°37′03.78″ | 1815 | Cemetery entrance, Cableway |

### 2.3.2. Chamois Ski Area

The territory at the top of the Chamois municipality is a ski area, but it may also be reached in the other seasons of the year on foot or by means of a chairlift that passes over the many paths that lead at a picnic area located near Lake Lod. Figure 5 shows, on a map, the position and the name given to the acoustic measurement points in this upper part of Chamois.

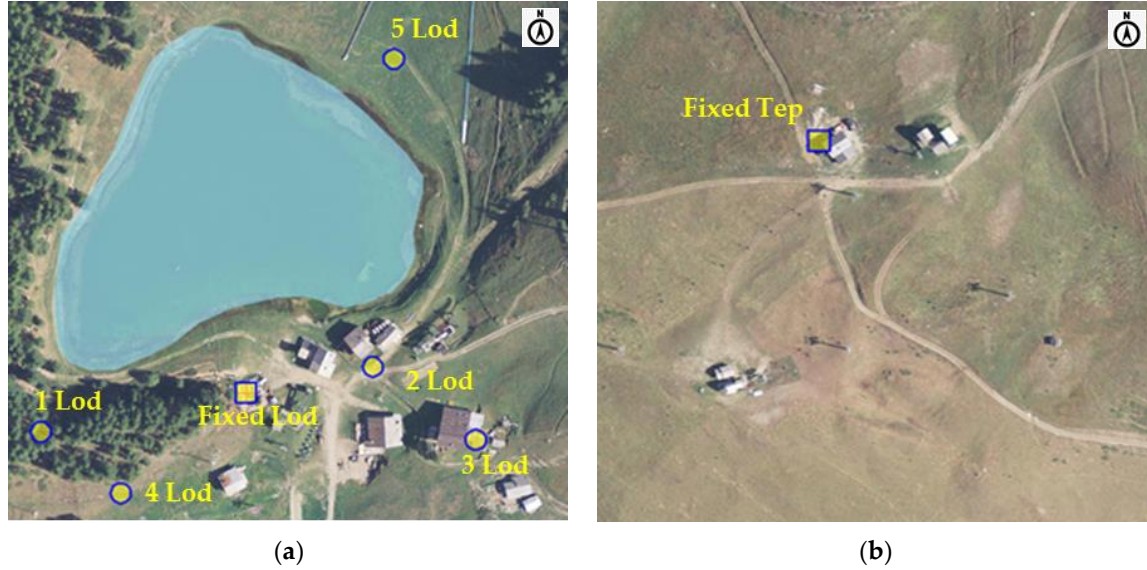

(**a**)　　　　　　　　　　　　　　　　　　　　　(**b**)

**Figure 5.** Picnic area at Lake Lod, with (**a**) the positions and the names given to the acoustic measurement points and (**b**) the Teppa location in the upper part of the ski area. As in Figure 4, the squares stand for fixed monitoring points and circles for the surrounding short-term monitoring points. Source: http://geonavsct.partout.it/pub/GeoCartoSCT/index.html.

The main characteristics of the territory are given in Table 3.

**Table 3.** Chamois ski area: details of the acoustic measurement locations.

| Site | Measurement Point | Geographic Coordinates (WGS84 N/E) | | Altitude (m a.s.l.) | Features of the Site |
|---|---|---|---|---|---|
| | Fixed Lod | 45°50′40.28″ | 007°37′22.24″ | 2030 | Arrival station of the chairlift from Chamois |
| | 1 Lod | 45°50′39.60″ | 007°37′17.52″ | 2030 | Picnic area next to the Lod Lake |
| | 2 Lod | 45°50′40.80″ | 007°37′17.52″ | 2025 | Area in the middle of the huts |
| Lod Lake ski area | 3 Lod | 45°50′39.60″ | 007°37′27.54″ | 2030 | Panoramic terrace bar overlooking Chamois |
| | 4 Lod | 45°50′38.64″ | 007°37′19.38″ | 2025 | Courtyard of a hut near the chairlift |
| | 5 Lod | 45°50′45.72″ | 007°37′25.50″ | 2020 | Next moving walkways near lake |
| Teppa ski area | Fixed Tep | 45°50′46.14″ | 007°38′00.62″ | 2250 | Chairlift at an intermediate station |

### 2.3.3. Cheneil Area

The Cheneil area extends from the small village of Cheneil in the plain at the foot of a valley, which is characterized by woodlands and pastures, to the top of the mountains above. The measurements were carried out around the village houses (Figure 6).

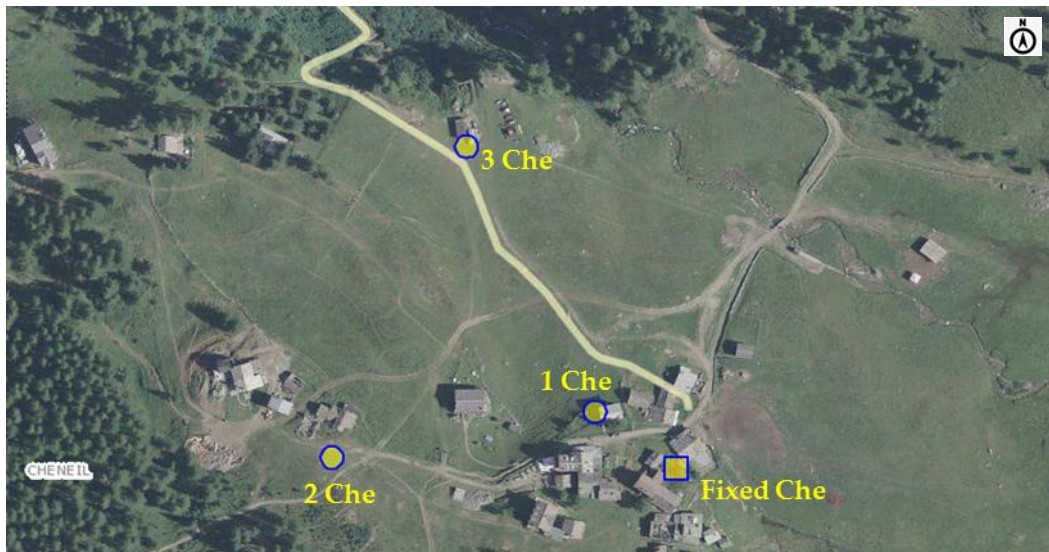

**Figure 6.** Acoustic measurement points in the Cheneil valley: one for fixed monitoring (square) and the others for short-term monitoring (circles). Source: http://geonavsct.partout.it/pub/GeoCartoSCT/index.html.

The main territorial characteristics of the territory are given in Table 4.

**Table 4.** The Cheneil village and its surrounding area: details of the acoustic measurement locations.

| Site | Measurement Point | Geographic Coordinates (WGS84 N/E) | | Altitude (m a.s.l.) | Features of the Site |
|---|---|---|---|---|---|
| Cheneil village and surrounding pastures | Fixed Che | 45°51′51.21″ | 007°38′38.36″ | 2100 | Courtyard in the "Bich"Alpine Refuge |
| | 1 Che | 45°51′51.96″ | 007°38′36.72″ | 2095 | Courtyard in a hut near the Refuge |
| | 2 Che | 45°51′51.30″ | 007°38′31.44″ | 2110 | "Cheneil les Gorret" dirt road |
| | 3 Che | 45°51′55.92″ | 007°38′34.14″ | 2090 | Arrival station of the funicular from the car park |

### 2.3.4. The Breuil-Cervinia Area

Breuil-Cervinia is one of the highest ski resorts in Europe, which means low temperatures and consistent snow falls in winter [14]. Skiing is practised nearly all year-round, because of the presence of a glacier at the top of the resort. Six fixed monitoring points were set up in Breuil-Cervinia: 3 of these points were located in the urban center and 3 in the ski area (at Plan Maison, at Cime Bianche and at Plateau Rosà). As for the other resorts, 3-4 stations were set up in the Breuil-Cervinia village (Figure 7) for short-term measurements, the characteristics of which are shown in Table 5.

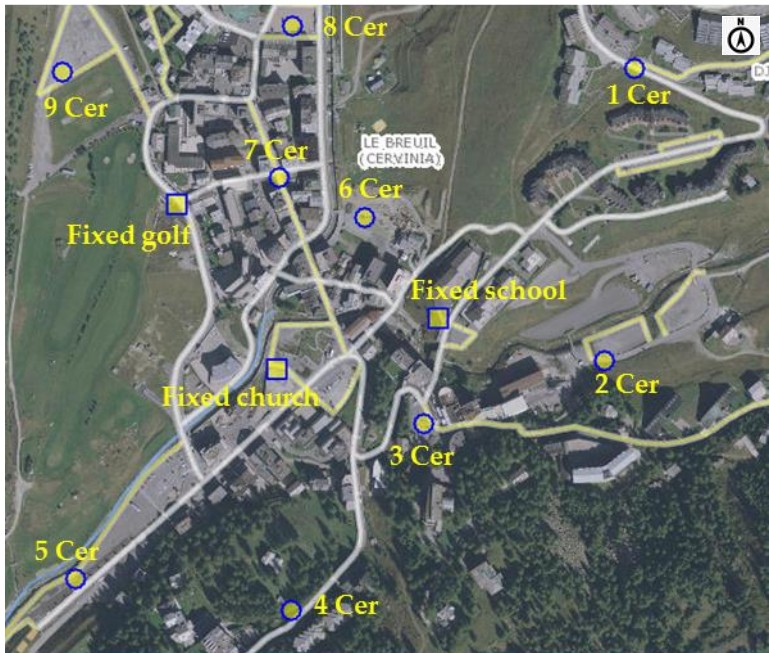

**Figure 7.** Acoustic measurement points in the Breuil-Cervinia village. The squares represent the three fixed points and the circles the surrounding short-term monitoring points. Source: http://geonavsct.partout.it/pub/GeoCartoSCT/index.html.

**Table 5.** The Breuil-Cervinia area: details of the measurement sites.

| Site | Measurement Point | Geographic Coordinates (WGS84 N/E) | | Altitude (m a.s.l.) | Features of the Site |
|------|-------------------|-------------------------|---|---------------------|----------------------|
| high area | Fixed at school | 45°56′05.55″ | 007°37′56.36″ | 2060 | terrace of School |
| | 1 Cer | 45°56′14.34″ | 007°38′06.54″ | 2085 | Breuil high part |
| | 2 Cer | 45°56′04.08″ | 007°38′04.92″ | 2055 | garden under cableway start |
| | 3 Cer | 45°56′01.92″ | 007°37′55.50″ | 2030 | ticket office square |
| central area | Fixed at square church | 45°56′03.58″ | 007°37′48.70″ | 2005 | near Cervinia Church |
| | 4 Cer | 45°55′55.02″ | 007°37′49.26″ | 2040 | area in front Hotel |
| | 5 Cer | 45°55′56.04″ | 007°37′38.46″ | 2000 | square at village entrance |
| | 6 Cer | 45°56′09.78″ | 007°37′52.68″ | 2015 | area near ice skating rink |
| suburb area | Fixed near golf course | 45°56′09.19″ | 007°37′43.13″ | 2015 | near Cervinia golf course |
| | 7 Cer | 45°56′10.32″ | 007°37′48.42″ | 2010 | the Cervinia main street |
| | 8 Cer | 45°56′16.14″ | 007°37′48.36″ | 2015 | car park at the end of the main street |
| | 9 Cer | 45°56′13.56″ | 007°37′36.60″ | 2015 | another car park |

### 2.3.5. The Breuil-Cervinia Ski Area

The Plan Maison area, at 2500 m a.s.l (Figure 8), was chosen to set up the short-term measurements near the fixed points in the ski area, which extends from the village center, at 2000 m a.s.l., to the Plateau Rosà glacier, at 3500 m a.s.l. No short-term measurements were conducted in the highest part of the ski area because the two long-term measurement points at the Cime Bianche Lake and at Plateau Rosà were considered to be representative of the entire ski area (Figure 9).

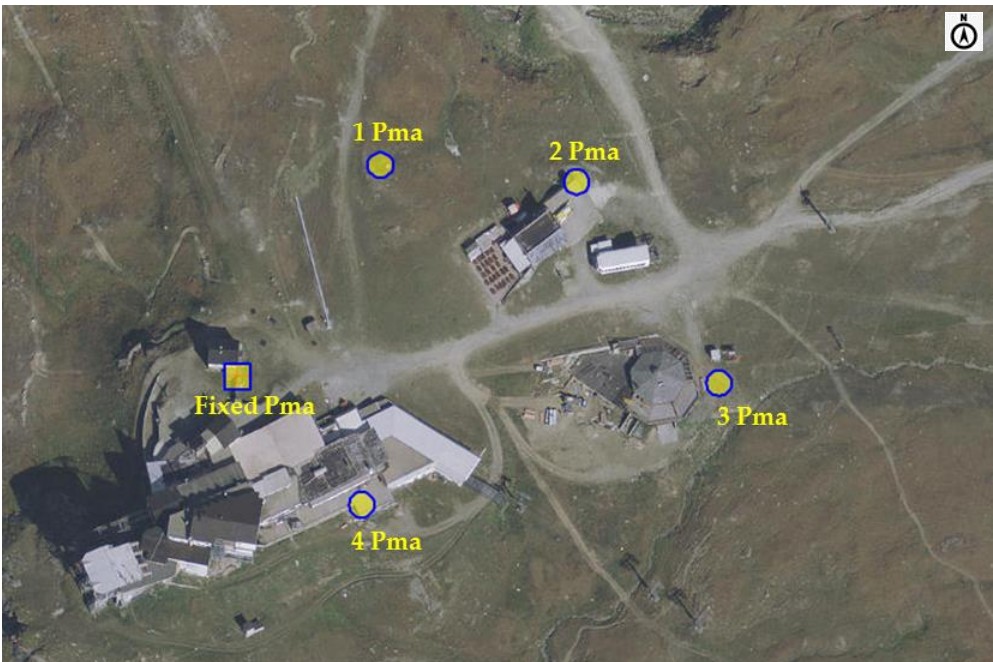

**Figure 8. The** Plan Maison area: position and the name given to the acoustic measurement points. Source: http://geonavsct.partout.it/pub/GeoCartoSCT/index.html.

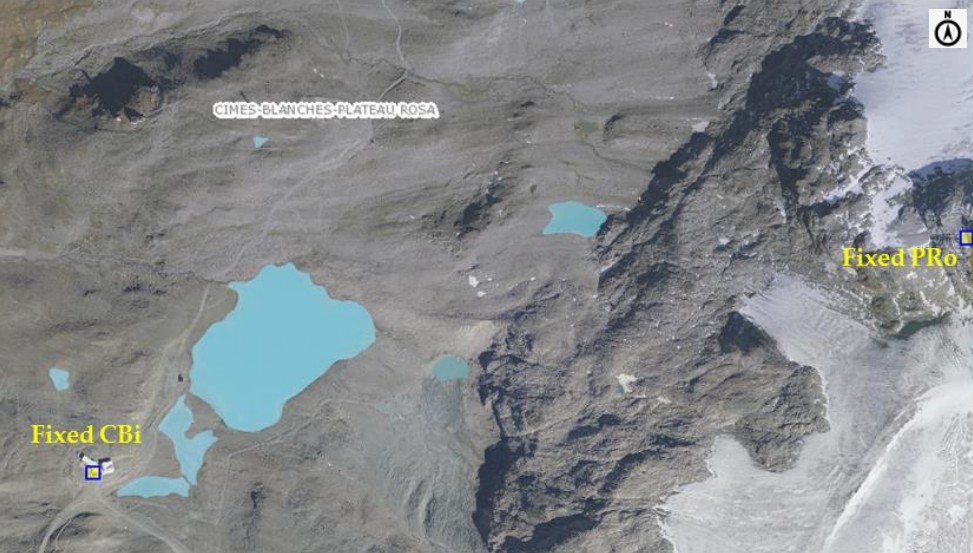

**Figure 9.** The Cime Bianche Lake and Plateau Rosà long-term measurement sites in the higher part of the ski area. Source: http://geonavsct.partout.it/pub/GeoCartoSCT/index.html.

Table 6 summarizes the geographical characteristics of the stations within the ski area.

**Table 6.** The Breuil-Cervinia ski area: details of the location of the measurement points.

| Site | Measurement Point | Geographic Coordinates (WGS84 N/E) | | Altitude (m a.s.l.) | Features of the Site |
|---|---|---|---|---|---|
| The Plan Maison ski area | Fixed Pma | 45°56′28.83″ | 007°39′16.55″ | 2550 | Terrace of a cableway station |
| | 1 Pma | 45°56′31.56″ | 007°39′19.08″ | 2540 | Along the ski run to Cervinia |
| | 2 Pma | 45°56′31.44″ | 007°39′22.68″ | 2555 | Terrace in front of the chairlift |
| | 3 Pma | 45°56′28.86″ | 007°39′25.32″ | 2545 | Area near the Plan Maison ski lift |
| | 4 Pma | 45°56′27.24″ | 007°39′18.84″ | 2550 | Terrace near a restaurant |
| The Cime Bianche ski area | Fixed CBi | 45°55′47.50″ | 007°40′49.12″ | 2815 | Intermediate cableway station |
| The Plateau Rosà ski area | Fixed PRo | 45°56′06.72″ | 007°42′26.27″ | 3460 | Terrace of the final cableway station |

## 2.4. Acoustic Classifications of the Municipalities with Regard to the Territorial and Infrastructure Characteristics

Italian legislation [15] has established criteria that should be adopted to subdivide municipal territories into acoustic classes on the basis of: the actual use of the territory, knowledge of its morphology, environmental and territorial planning, the road and transportation networks, and the presence of industrial production activities (Table 7).

**Table 7.** Criteria provided by national law for the classification of municipal territories.

| Acoustic Class | Name | Characterization |
|---|---|---|
| Class I | Protected areas | Quietness represents a basic element for the use of such areas: hospital, schools, rest and leisure areas, rural residential areas, areas of particular urban interest, public parks, etc. |
| Class II | Residential use areas | Urban areas that are affected by local vehicular traffic, with a low population density, a limited presence of commercial activities, and the absence of industrial and craft activities |
| Class III | Mixed type areas | Urban areas affected by local or passing through traffic with an average population density, the presence of commercial activities, offices, a limited presence of craft activities, no industrial activities, and rural areas with agricultural machines |
| Class IV | Intense human activity areas | Urban areas affected by intense vehicular traffic, with a high population density and a high presence of commercial activities and offices, the presence of craft activities, areas close to major roads and railway lines, areas with small industries |
| Class V | Prevalently industrial areas | Areas affected by industrial settlements with just a few dwellings |
| Class VI | Purely industrial areas | Areas that are affected exclusively by industrial activities and where there are no dwellings |

Following the drawing up of the national legislation on acoustic classes, Regional Council resolution no. 2083/2012 [16] considered the preservation of quietness in open countryside areas, and in places generally far from an urbanized context and not affected by the main infrastructures or human activities, through the introduction of "class 0", in addition to the classes foreseen by the national law. Class 0 provides indications on the zoning of particular areas where specific human activities take place (e.g., mountain pastures, mountain huts, ski resorts) with the aim of preserving the integrity of the acoustic climate in such areas, which are situated far from sources of noise from either mountain pasture activities (electric generators, chain saws, power grass-mowers) or from tourist-sporting activities, such as the electric motors of cableways and/or ski lifts, and artificial snow guns.

For the first time, a geometrical criterion of the distance from artificial sources has been used to subdivide the different types of areas; furthermore, the seasonal nature of any production activities that

may take place within the mountain areas in which skiing infrastructures are present was considered for both the winter and summer seasons.

Table 8 shows the criteria provided by the Aosta Valley regional law for specific mountain activities and infrastructures, including the ski resorts and their installations, for which the application of different limit values in the winter season is permitted.

**Table 8.** Criteria provided by the Aosta Valley regional law for specific mountain activities and infrastructures.

| Type of Activity or Infrastructure | Specific Acoustic Classification | Extension Area (m) |
|---|---|---|
| Mountain farm pastures | Should be classified as class III for a radius of 100 m surrounding a farmhouse | Class II for 100 to 200 m, and class I for a pasture |
| Mountain huts | Should be classified as class II | with 100 m class I surrounding |
| Bars and restaurants | Their immediate surrounding (déhors and external appliances) can be included in class III | in class II for further 100 m and in class I from 100 to 200 m of distance |
| Cableway stations, ski lifts, and chairlifts | Should be classified as class IV in the periods they are open | to be extended to the relevant property areas |
| Ski slopes | Should be classified as class IV during the skiing activity period (but as a lower class in the other periods, depending on the use of the territory and the soundscape) | must be provided around the slopes and not exceeding 150 m |
| Remote high mountain areas | Class 0 Areas where the environmental sound levels must not be raised from artificial sources | There is no limit to the extension area around these areas, but it must border on with class I areas |

Therefore, a specific acoustic classification was applied for the winter season at the ski resorts. Figure 10 shows the seasonal acoustic classification of the two studied sites involved in downhill skiing (Breuil-Cervinia and Chamois).

There are no ski lifts in the Cheneil valley and only ski mountaineering and snow racket trekking are practiced in winter. Since this type of attendance does not require a seasonal acoustic classification (Figure 11), the entire area was included in the two most protective acoustic classes throughout the year. Hikers and mountain bikers frequent the valley in summer, while the pastures are used for grazing cows and sheep. In these sites, which are generally far from urbanized contexts where the main infrastructures and human activities are most prevalent, the environment has an unspoiled acoustic climate.

This means that most of the Aosta Valley territory is part of class 1 (56%) or class 0 (15%), which is specifically required in regional regulations to preserve the naturalness of remote, high mountain areas that generally extend above 2500 m of altitude. Twenty percent of the remaining territory falls into class 2, while only 9% falls into the third to the sixth classes (Figure 12). Most of the Gran Paradiso National Park and the Mont Avic Regional Park territory and most of the main protected areas in the Aosta Valley region also fall into the first two acoustic classes (0 and 1).

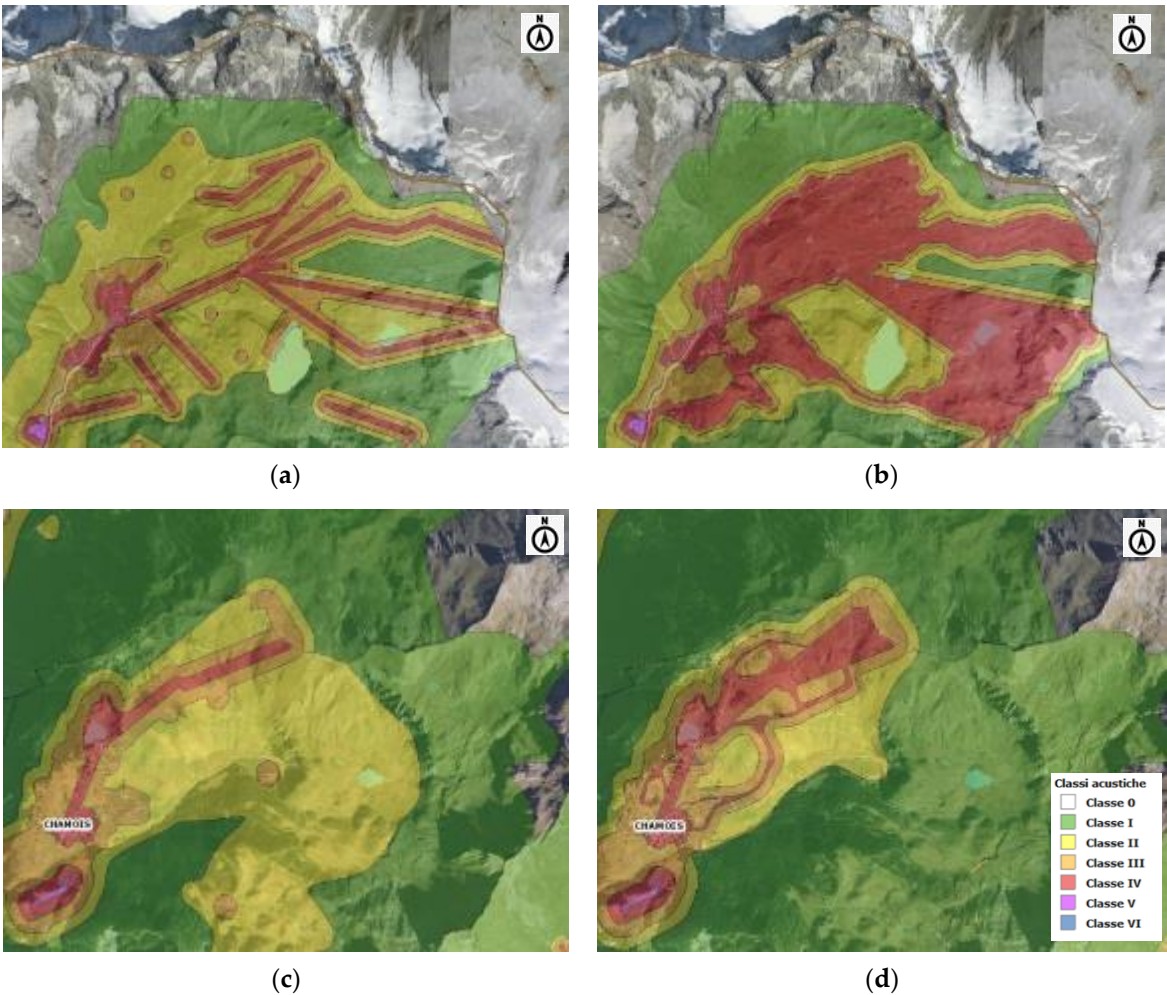

**Figure 10.** Acoustic classification—(**a**) the Breuil-Cervinia ski area; (**b**) the Breuil-Cervinia ski area in the winter season; (**c**) the Chamois ski area; (**d**) the Chamois ski area in the winter season. Source: http://geonavsct.partout.it/pub/GeoCartoSCT/index.html.

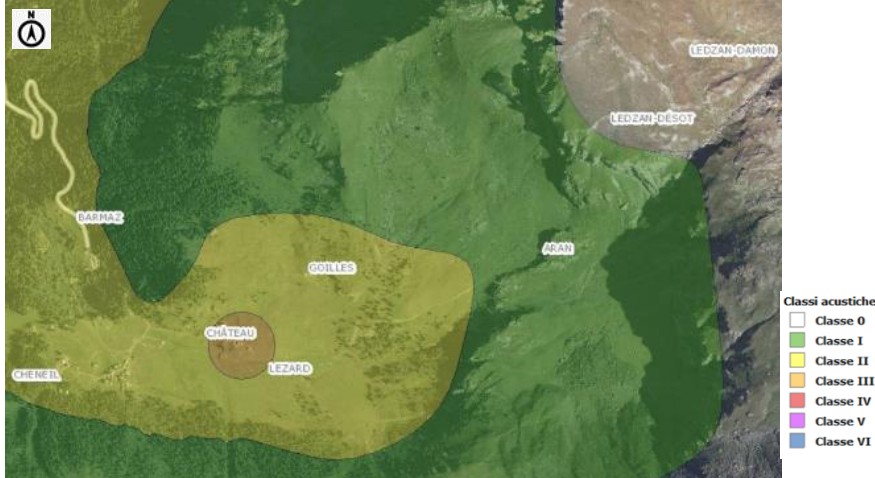

**Figure 11.** Cheneil resort: the acoustic classification is the same all year-round. Source: http://geonavsct.partout.it/pub/GeoCartoSCT/index.html.

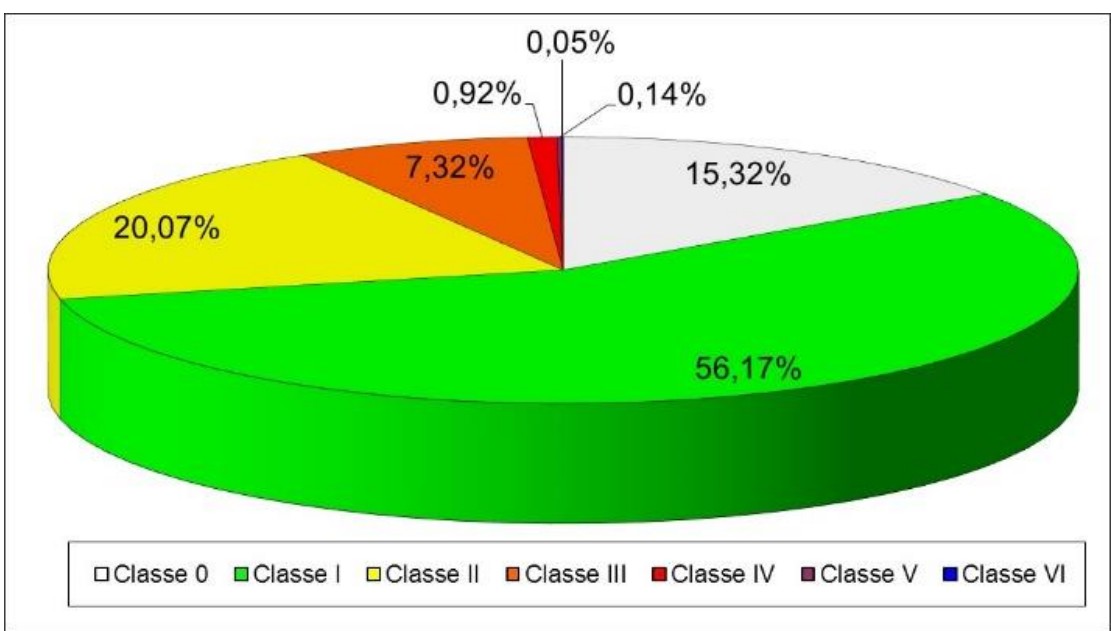

**Figure 12.** Percentage of regional territory in each of the seven acoustic classes.

*2.5. The Case Study: Measurement Methodology and Data Processing*

Since the three different tourist resorts in Valtournenche that were selected for the study can have different types of seasonal soundscapes, it is important to understand how to preserve the natural acoustic climate in each resort. The problem can be resolved by considering the acoustic climate as an integral part of the mountain environment, with an integrated management of the tourism development, taking into account both the need for leisure and entertainment and the right to rest and quietness [17].

2.5.1. Measurement Methodology

Long-term phonometric measurements were carried out in each of the chosen areas for a few days (with an acquisition frequency of 1 sample per second) in 3 different seasons of the year: in summer, in the month of August; in autumn, in the months of September and October; and in winter, in the months of December and January. The microphone was positioned at a variable height (from about 2 to 4 m), depending on the site, but maintained at the same height for the 3 different periods of the year.

At the same time, short-term and assisted measurements, lasting 10 min, were carried out on four time bands of the day to cover the entire day [18,19]. In order to best characterize the acoustic climate, at least four intervals were chosen for the daytime period and two intervals for the nighttime: 9–12, 14–17, 17–20, and 22–24. A number of short-term measurement stations, ranging from 3 to 5, were set up around each of the fixed ones to confirm its representativeness and to obtain a greater territorial monitoring coverage. Short-term measurements were made using a microphone at a height of 1.5 m above the ground, with a time history sampling time of 0.1 s, and an audio file was recorded.

The main acoustic parameters used for the evaluation of the sound climate (LAeq and percentile levels) were acquired in all the measurements. Short-term measurements were not carried out in the most remote sites of the considered ski areas in Breuil-Cervinia (Plateau Rosà and Cime Bianche) and Chamois (Teppa), because the long-term measurement points were considered to be representative, from an acoustic point of view, of the entire surrounding area (Table 9).

2.5.2. Meteorological Conditions During the Measurements and Data Validation

Meteorological data, provided by the Regional Functional Centre (CF VdA) [20] and by the Research Centre on the Energy System (OASI-RSE) [21], were analyzed to validate the acoustic levels

acquired in the long-term measurements. Data affected by rainfall were discarded, as were those with wind speeds exceeding 5 m/s. Figure 13 shows a map with indications of the weather stations that provided the meteorological data in Valtournenche.

**Table 9.** Number of measurements, the total hours of the measurements and the characteristics of each measurement site.

| Municipality | Site (m a.s.l.) | No. Long-Term Measurements | Hours of Long-Term Measurements | No. Short-Term Measurements | Hours of Short-Term Measurements |
|---|---|---|---|---|---|
| Chamois | Village (1819) | 3 | 120 | 48 | 8 |
| | Lake Lod ski area (2029) | 3 | 168 | 27 | 5 |
| | Teppa high ski area (2248) | 3 | 180 | / | / |
| Valtournenche | Urban center (2010) | 9 | 396 | 108 | 18 |
| | Plan Maison station in middle ski area (2550) | 3 | 284 | 23 | 4 |
| | Cime Bianche high ski area (2820) | 3 | 308 | / | / |
| | Plateau Rosà station on the glacier (3460) | 3 | 180 | / | / |
| | Cheneil rural village and pasture (2101) | 3 | 55 | 35 | 6 |
| | Total | 30 | 1691 | 241 | 41 |

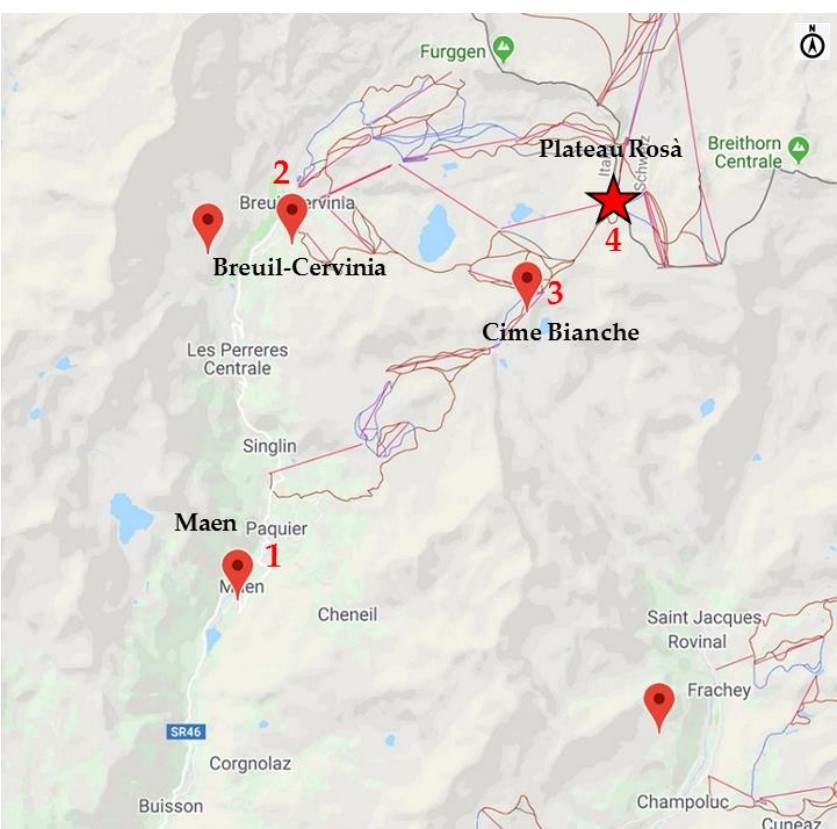

**Figure 13.** Valtournenche—Map of the weather stations where the data were obtained; the pointer symbol indicates the CF VdA stations and star symbol the OASI-RSE station. Source: http://geonavsct. partout.it/pub/GeoCartoSCT/index.html.

The short-term measurements were carried out in the absence of atmospheric precipitation (rain or snow) and of strong winds under mostly clear skies. It should be noted that there was an abundant amount of snow on the ground in the high elevation areas in the winter period.

*2.6. The Study: Data Processing*

The several measurements that were carried out and the large amount of data provided by the different sound level meters oriented the data elaboration towards a standard archiving and presentation format. To do this, the data were analyzed using Excel worksheets.

2.6.1. Homogenization of the Data for Processing

Data from both the long-term and short-term measurements were processed, starting from original text files and paged in the same way in the Excel sheets using the Visual Basic for Applications code (VBA). Three charts were produced with an LAeq time history, 1/3 octave band spectra and a statistical distribution of the percentile levels for each short-term measurement performed with a sampling time of 0.1 s.

The long-term measurement data are processed in the same way, but only considering the LAeq time history chart with a sampling time of 5 s, in order to compare the sound pressure level trends on the same graph over a 24-h period for the different measurement seasons.

All these analyses were carried out for each short-term measurement; the results are available at the Aosta Valley Regional Environmental Protection Agency (ARPA). The entire script that allowed the automatic compilation of the Excel sheets from the measurement file can be found in Annex 13 of the text of the complete study at the following address https://webthesis.biblio.polito.it/7856/.

2.6.2. Data Analysis Based on the Main Detected Acoustic Parameters

The measurements that were carried out made it possible to obtain sound levels that clearly represent the changes in the seasonal acoustic climate in the studied rural and mountain areas. A graphic comparison of the 24-h time histories and a comparison of the acoustic levels were in particular carried out for each relevant site and for the three different periods of the year (the summer high season in August, the autumn low season in September and October, and the winter season in December and January).

One of the first analyses involved carrying out a correlation between the sound pressure levels of the long-term measurements and those of the short-term ones conducted in the surrounding areas. This comparison confirmed the representativeness of the sound levels at the fixed measurement points for the surrounding area and it also allowed the sound levels to be evaluated over a larger area around the fixed point with the aim of conducting a possible future mapping.

All the levels acquired at each site were inserted into the same graph, without any seasonal or time distinctions, and the average LAeq and LA95 levels of each of the 4 time bands of the prolonged measurements were correlated with the respective LAeq and LA95 levels of the short-term measurements carried out in the surrounding area. The objective was to highlight the good correlation of the fixed-point levels with those on the larger surveyed area.

The two graphs in Figure 14 show, respectively, the correlation between the LAeq levels (Figure 14a) and the correlation between the LA95 percentile (Figure 14b) of one of the monitored locations. The levels refer to the same measurement time. A different symbol was chosen for each short-term measurement point in each monitored area.

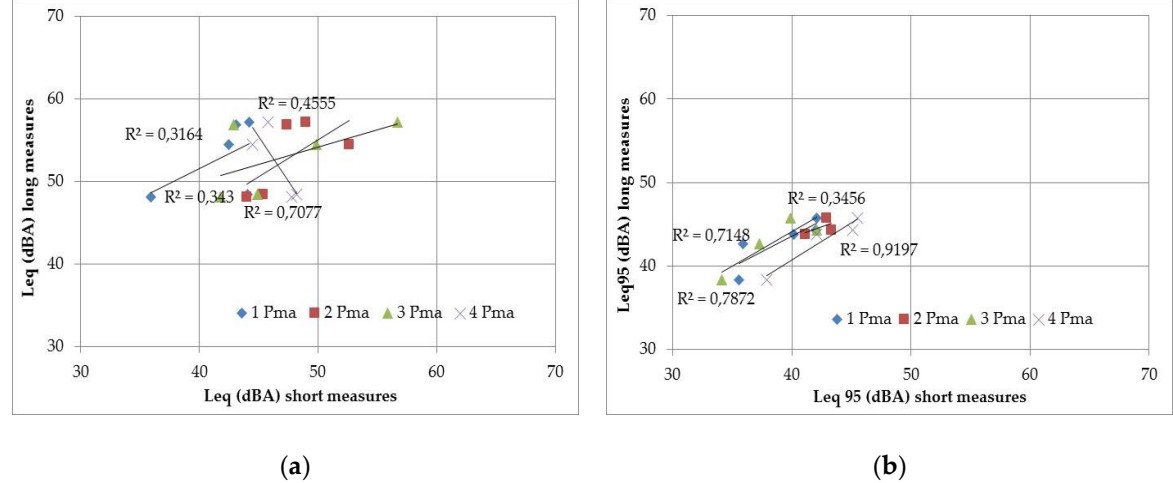

**Figure 14.** The Plan Maison ski area. Correlation between the long-term measurement results and the short-term measurement results for (**a**) the LAeq level and (**b**) the LAeq95 background level. Each symbol represents a short-term measurement point in the area (Table 6).

A better correlation was generally found for the background sound pressure level (LA95eq) than for the LAeq levels. This is due to individual but noisy events, which occur near the short-term measurement points, and influence LAeq more than LAeq95 (background level).

Another analysis was performed on the acoustic parameters to evaluate the changes in the acoustic climate during the three different periods of the year studied in the work, and the differences in the 24-h time histories were shown on a single graph. The results of this analysis are reported in Chapter 3.

### 2.7. The Harmonica Index Applied to the Measured Levels

Information about the sound climate is usually presented in the form of measurement results of the main acoustic indicators that have been used (LAeq, Ln, Lden...). However, these indicators are usually difficult to explain to people who are not specialized in the field. Furthermore, the decibel unit, used for these indicators, has the disadvantage of being expressed on a logarithmic scale rather than a linear one.

For these reasons, the HARMONICA project has proposed a new index, which is particularly innovative for three reasons:

- It takes into account both background sound pressure levels and peak levels due to events that occur on it and can easily be calculated from the LAeq and LA95eq parameters that all sound level meters can acquire.
- It can easily be understood by the general public because it is expressed as a number on a scale from 0 to 10, without the use of decibels, in order to make it easy to understand.
- It can be calculated for any time period (hourly, daily, monthly, etc.).

In this study, in order to reflect changes in the acoustic climate during the day, an hourly index, which allows a clear representation of the variation in sound levels, taking into account both the background and the peak levels that occur in the area, was calculated for each monitored site. Index values were also calculated for daytime periods (from 6:00 to 22:00) and for nightime ones (from 22:00 to 6:00).

The formula for the index was set by testing the various descriptors that were selected and validating them on a database of elementary acoustic measurements (LAeq,1s) from 24 sites that are representative of the eight main types of noise exposure (land transport noise, air traffic noise, and quiet areas) acquired by the Bruitparif noise observatory [22], in order to take into account the diversity of the local environmental situations. The tests led to the coefficients of the index formula being adjusted, in order to present the variations of the hourly sound levels on a scale from 0 to 10.

The mathematical definition of the harmonica index is:

$$I_H = BGN + EVT = 0.2 \times (LA95eq - 30) + 0.25 \times (LAeq - LA95eq)$$

The first part of the $I_H$ index relates to the background level and it has been named the BGN sub-index. The second part of the index refers to the event-related component and represents the acoustical energy provided by sound peaks that emerge above the background level; it is named the EVT sub-index. The $I_H$ index has a possible score of 0 to 10, rounded to one decimal place.

Classification of the Acoustic Quality of Different Areas and Comparison with the Threshold Value

In the present study, the harmonica index was applied, as defined in the relative project, without modifying the formula, in order to be able to compare its value with similar situations in Europe. Furthermore, in order to classify the acoustic environment quality of different areas, three colors (green/yellow/red) were associated with the index in order to relate the measured sound levels to the environmental quality objectives of the World Health Organisation (WHO) and to the values recognized as being critical for noise [23].

The color scale and the values of each class were defined taking into account the time of day, as people are more sensitive to noise at night, in the following way (Table 10):

- Green indicates that the noise level is below 4 during the day and below 3 at night, that is, a level of 45 dBA at which WHO states noise is likely to disturb sleep;
- Yellow shows the index is between 4 and 8 during the day, and between 3 and 7 at night;
- Red appears if the index is greater than or equal to 8 during the day, and 7 at night. These levels can be achieved for constant levels of 70 dBA and 65 dBA, respectively, which are widely recognized in Europe as being the critical thresholds for exposure to noise.

**Table 10.** Reference values for the acoustic quality classification of an environment through the $I_H$ index.

| Color of the Index $I_H$ | Quality Acoustic Environment | Day Period (06-22) | Night Period (22-06) |
|---|---|---|---|
| Red | Recognized by WHO for critical acoustic quality environments | $I_H \geq 8$ | $I_H \geq 7$ |
| Yellow | Environments where noise exceeds the recommended quality but remains under the recognized critical thresholds | $4 \leq I_H < 8$ | $3 \leq I_H < 7$ |
| Green | Good acoustic environments under the quality thresholds | $I_H < 4$ | $I_H < 3$ |

## 3. Results

### 3.1. Acoustic Characterisation of Different Seasons Based on the Phonometric Parameters

An example of a visual comparison of time histories is reported in the following two figures. In the first figure (Figure 15), the comparison refers to the sound levels measured in the three different seasons in the urban center of Breuil-Cervinia, while the second one (Figure 16) shows a comparison pertaining to the Plan Maison site, in the ski area near the intermediate cableway station that leads to Plateau Rosà.

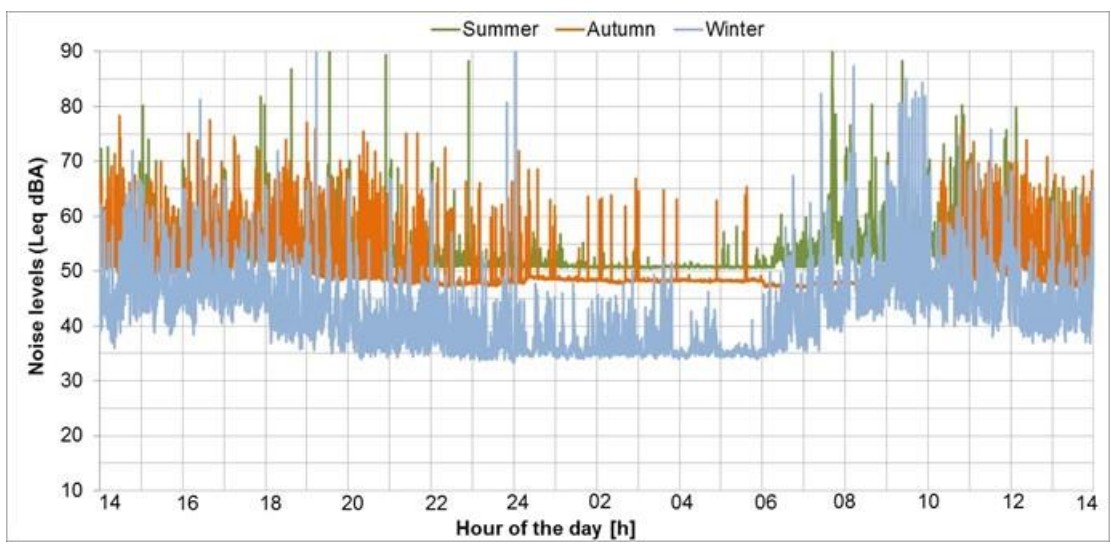

**Figure 15.** Breuil-Cervinia church square: comparison of the LAeq time histories over 24 h (summer, autumn, and winter).

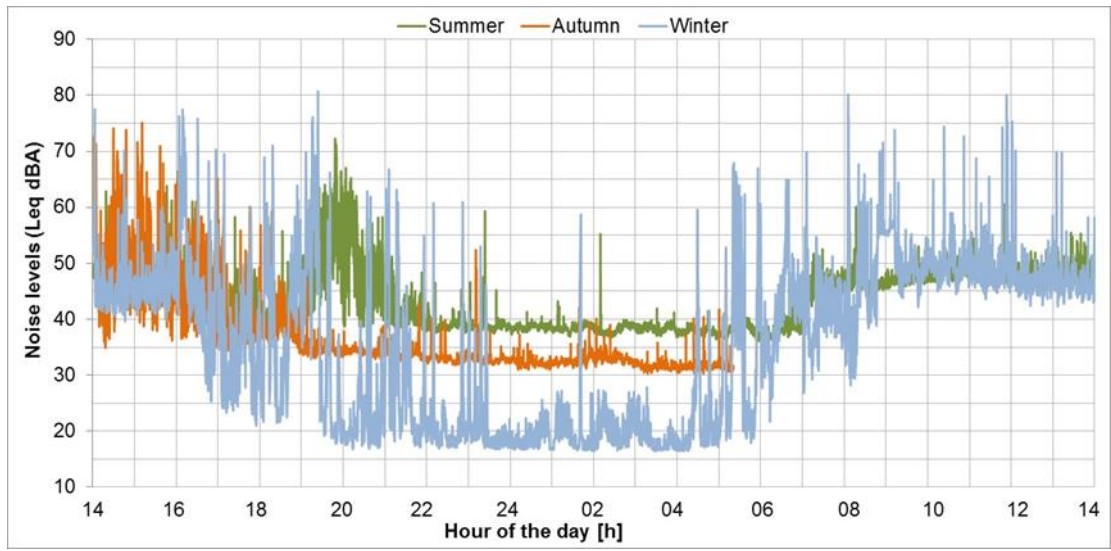

**Figure 16.** Plan Maison ski area: comparison of the LAeq time histories over 24 h (summer, autumn, and winter).

The comparison shows that the background sound levels in the urban center of Breuil-Cervinia are generally higher for all the seasons than in the intermediate area of Plan Maison, which is located within the ski area. This is due to the greater anthropization of the area, with the presence of local roads and the regional road that climbs from the bottom of the valley. There are more or less peaks during the daytime, depending on the season, mainly due to the presence of cars and the influx of tourists to the pedestrian commercial-reception area.

On the other hand, there is a more evident variation in the sound levels in the medium- and high-altitude sites, which are mountain areas with predominantly quiet characteristics for half of the year, when the lifts are in operation (approximately 8:30 to 17:00). In these situations, the peak levels are comparable with those measured in the countryside and are mostly due to the presence of humans, while the background levels are increased by the sound emissions from the lifts in winter and by the presence of streams, birds, and insects in summer. In winter, the tracks show higher equivalent levels, due to the maintenance operations of the ski slopes, starting from a few hours before the ski lifts open and finishing a few hours after they close [24].

The same is true for the center of the village of Chamois, where, despite the absence of cars, the equivalent sound pressure level in summer and autumn is influenced by the numerous agricultural and land maintenance activities.

All the sound levels of each seasonal time history were extracted. Summaries of all the LAeq and LA95eq level values, subdivided into the two time periods foreseen by the national legislation (daytime: from 6:00 to 22:00 and nighttime: from 22:00 to 6:00), are provided in Tables 11 and 12, respectively.

As already highlighted from the comparison of the trends of the time histories, the sound levels reported in the above tables also show that the background sound level is lower in winter (sometimes even 20 dBA lower) than in the other two considered seasons: autumn and summer. This is certainly due to the fact that the fountains and streams in the area have less water in winter and the snow on the ground dampens almost all the natural sources of sound. There is also a lack of natural sound from birds and insects, which increases the background sound level in summer. Generally, the equivalent daily levels in the areas most frequented by tourists (those used for sports) are still higher than 50 dBA, even where there are no cars: people's voices and the barking of dogs, together with pastoral activities, are probably not negligible sound sources. This was particularly evident in the measurements carried out in the Cheneil valley, where although the locality has no fixed machinery or artificial plants, the equivalent level was influenced to a great extent by the barking of dogs in the area [25].

**Table 11.** LAeq (dBA) detected in the acoustic measurement campaign divided by the period of the day and by the seasons.

| Site | Daytime (06-22) | | | Nighttime (22-06) | | |
|---|---|---|---|---|---|---|
| | Summer | Autumn | Winter | Summer | Autumn | Winter |
| Chamois village | 57.2 | 55.7 | 53.5 | 50.1 | 48.8 | 46.2 |
| Chamois-Lake Lod in the ski area | 54.0 | 36.6 | 53.9 | 49.7 | 29.5 | 28.3 |
| Chamois-Teppa, high area in the ski area | 38.7 | 31.5 | 48.8 | 25.9 | 23.7 | 22.7 |
| Cervinia-church square | 63.6 | 54.7 | 56.6 | 51.5 | 49.4 | 53.8 |
| Cervinia-Giomein place | 58.4 | 52.4 | 55.9 | 52.4 | 48.9 | 44.0 |
| Cervinia-main road | 58.0 | 54.9 | 55.9 | 47.0 | 42.4 | 44.8 |
| Plan Maison station in the middle of ski area | 47.5 | 49.8 | 54.6 | 38.5 | 32.3 | 44.5 |
| Cime Bianche-high ski area | 59.7 | 36.8 | 63.1 | 35.0 | 33.7 | 50.0 |
| Plateau Rosà station on the glacier | 49.0 | 55.4 | 64.5 | 40.3 | 57.0 | 47.9 |
| Cheneil-rural village and pastures | 42.5 | 53.2 | 35.7 | 51.2 | 45.5 | n.r. |

**Table 12.** LAeq95 (dBA) detected in the acoustic measurement campaign divided by the period of the day and by the seasons.

| Site | Day Period (06-22) | | | Night Period (22-06) | | |
|---|---|---|---|---|---|---|
| | Summer | Autumn | Winter | Summer | Autumn | Winter |
| Chamois village | 50.8 | 47.5 | 44.3 | 48.4 | 47.3 | 44.3 |
| Chamois-Lake Lod in the ski area | 48.3 | 28.6 | 43.2 | 46.8 | 27.2 | 23.8 |
| Chamois-Teppa high ski area | 36.0 | 18.0 | 37.8 | 19.0 | 18.0 | 18.0 |
| Cervinia-church square | 51.3 | 48.3 | 40.0 | 50.1 | 48.0 | 34.2 |
| Cervinia-Giomein place | 52.0 | 47.5 | 40.5 | 51.3 | 47.4 | 38.1 |
| Cervinia-main road | 46.0 | 40.8 | 37.5 | 43.4 | 40.3 | 27.5 |
| Plan Maison station in the middle of ski area | 43.9 | 36.5 | 46.8 | 37.2 | 31.0 | 22.2 |
| Cime Bianche-high ski area | 45.6 | 30.9 | 50.2 | 32.8 | 29.7 | 35.3 |
| Plateau Rosà station on the glacier | 40.3 | 33.4 | 46.1 | 34.4 | 23.6 | 36.4 |
| Cheneil-rural village and pastures | 38.4 | 33.8 | 20.0 | 39.1 | 33.5 | n.r. |

*3.2. The Harmonica Index Calculated from the Measured Levels in the Differents Sites*

The harmonica index was quantified for the same 24-h day already considered in the previous calculations in order to offer a better understanding to non-expert users. The hourly trend and the values of the daytime and nighttime periods of the $I_H$ index were calculated for each area.

A histogram, divided into two parts, was chosen for the graphical representation of the harmonica index, in order to simultaneously provide several pieces of information about the acoustic environment in a concise and clear manner (see Figure 17): the blue color at the bottom of the bar represents the component related to the background level (BGN), while the red color at the top of the bar represents the event-based component (EVT) related to the sound peaks that appear above it.

In this paper, the results of the hourly index quantification are reported for the same two locations, that is, the urban center of Breuil-Cervinia and Plan Maison (in the ski area near the intermediate cableway station) which were previously considered to compare the 24 h time histories. The results concerning all the points can be found online at the following address https://webthesis.biblio.polito.it/7856/.

The $I_H$ index in the church square in the center of the Breuil-Cervinia village shows, in both summer and autumn, a larger share from background sound emissions (BGN), mostly due to the water flowing in the Marmore stream. In winter, the index is dominated by the event component (EVT), which, in that period, was produced by the clearing of the abundant snow that had fallen in the days preceding the acoustic measurements. The background sound environment is lower in this season than in the other two seasons.

For the Plan Maison ski area, the $I_H$ index shows a background level (BGN) of around 3 in summer and winter when the cableway that carries the tourists to the area is in operation (approximately 8:00 to 17:00), but which is much lower in the other periods of the day. However, it should be noted that some hourly data are missing (from 6:00 to 12:00) in the graphs of the autumn period for this site, which was caused by a malfunctioning of the instrumentation. In winter, the event component (EVT) contributes more to the index; this period of the year is in fact characterized by the presence of many tourists and by ski slope maintenance activities.

The "e" and "f" graphs in Figure 17 also suffer from a lack of the background noise (BGN) in the IH index in some parts. This occurred when background sound levels below 30 dBA were acquired, a situation in which the mathematical expression would have led to negative values of the index. Following a comparison, and on the suggestion of the harmonica project developers, it was decided to set this factor to zero, thereby avoiding negative values while preserving the original philosophy of the index. In these cases, only the part due to the events (EVT) is represented on the graph.

According to the project developers, such a condition represents an ideal background sound level, below which it is not necessary to acquire any quantitative information, because of the lack of noise-related health effects, in any context, for values below 30 dBA. This condition is always encountered at nighttime in the winter season in this type of mountain area, due to the absence of natural and animal sounds.

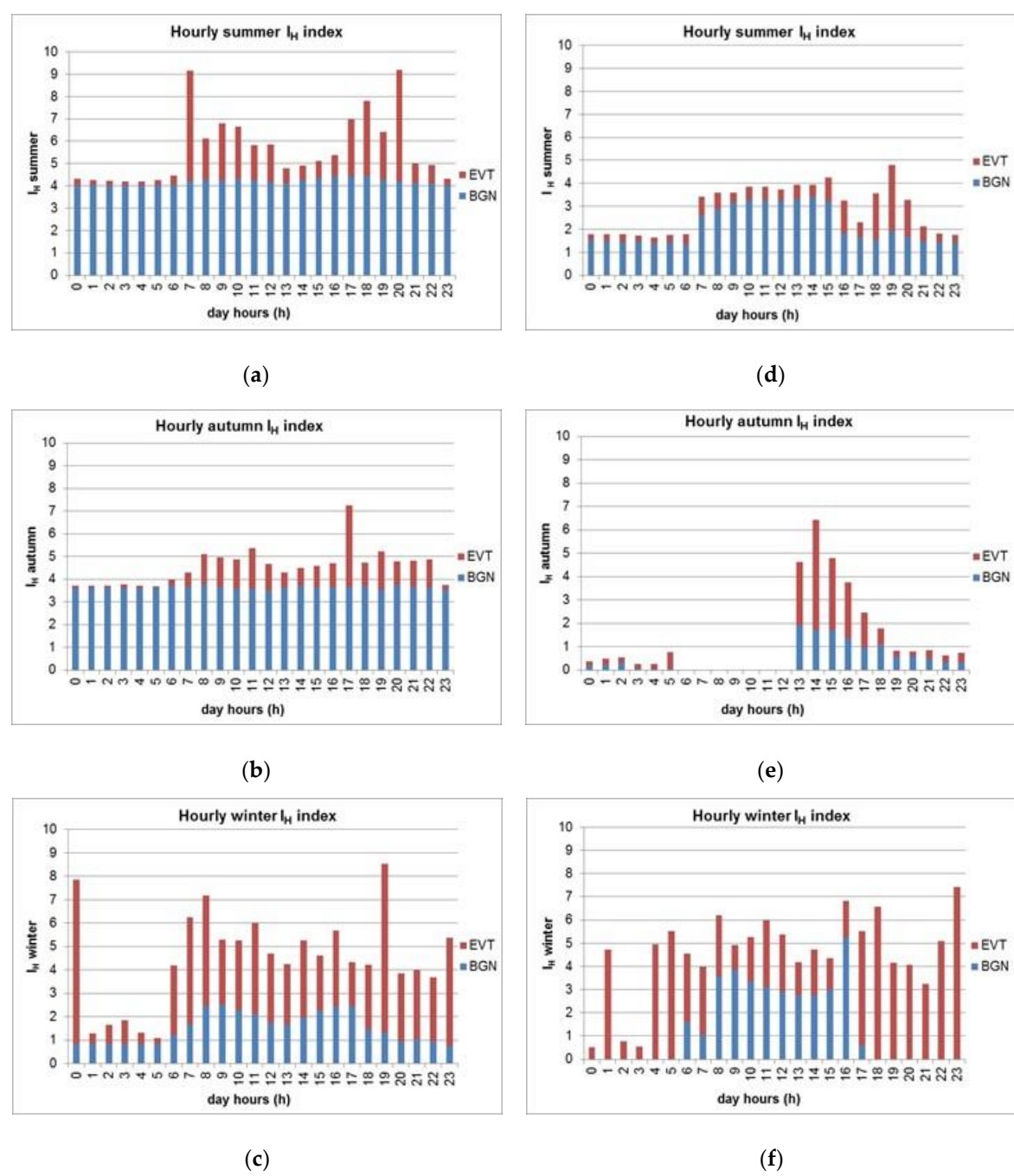

**Figure 17.** Hourly harmonica index ($I_H$) evaluated in the Breuil-Cervinia church square in (**a**) summer, (**b**) autumn, (**c**) winter, and at Plan Maison in (**d**) summer, (**e**) autumn, and (**f**) winter.

## 4. Discussion

In this study, in addition to collecting acoustic data in mountain resorts, an acoustic quality [26] factor, based on the value assumed by the $I_H$ index, has been associated to each area. The index has been calculated for the two reference times defined in Italian law: daytime, from 6:00 to 22:00, and nighttime, from 22:00 to 6:00.

The quality indicator was assigned at each measurement point on the basis of the relevant color, according to the $I_H$ index in Table 10, in Section 2.7. The results of the evaluation of the $I_H$ index for the day period are given in Table 13.

**Table 13.** The daytime $I_H$ index (06-22 h) for the three seasons for all the monitored resorts.

| Site | Day Period (06-22) | | | | | | | | |
| | Summer | | | Autumn | | | Winter | | |
| | bgn | evt | $I_H$ | bgn | evt | $I_H$ | bgn | evt | $I_H$ |
|---|---|---|---|---|---|---|---|---|---|
| Chamois village | 4.2 | 1.6 | 5.8 | 3.5 | 2.1 | 5.6 | 2.9 | 2.3 | 5.2 |
| Chamois-Lake Lod ski area | 3.7 | 1.4 | 5.1 | 0.0 | 2.0 | 2.0 | 2.6 | 2.7 | 5.3 |
| Chamois-Teppa high ski area | 1.2 | 0.7 | 1.9 | 0.0 | 3.4 | 3.4 | 1.6 | 2.8 | 4.4 |
| Cervinia-church square | 4.3 | 3.1 | 7.4 | 3.7 | 1.6 | 5.3 | 2.0 | 4.2 | 6.2 |
| Cervinia-Giomein place | 4.4 | 1.6 | 6.0 | 3.5 | 1.2 | 4.7 | 2.1 | 3.8 | 5.9 |
| Cervinia-ringroad | 3.2 | 3.0 | 6.2 | 2.2 | 3.5 | 5.7 | 1.5 | 4.6 | 6.1 |
| Plan Maison-middle station ski area | 2.8 | 0.9 | 3.7 | 1.3 | 3.3 | 4.6 | 3.4 | 2.0 | 5.4 |
| Cime Bianche-high ski area | 3.1 | 3.5 | 6.6 | 0.2 | 1.5 | 1.7 | 4.0 | 3.2 | 7.2 |
| Plateau Rosà station on the glacier | 2.1 | 2.2 | 4.3 | 0.8 | 4.9 | 5.7 | 2.6 | 5.2 | 7.8 |
| Cheneil-rural village and pastures | 1.7 | 1.0 | 2.7 | 0.8 | 4.9 | 5.7 | 0.0 | 3.9 | 3.9 |

The measurement areas in the tourist resorts where there are more buildings (Breuil-Cervinia and Chamois) have higher values of the $I_H$ index, although they generally do not exceed the average acoustic environment quality range. Only in three cases does the day index take on values close to 8, and it only exceeds this value in one case. The three cases with an index close to a value of 8 refer to situations of high tourist attendance (Breuil-Cervinia in the month of august and its ski areas in winter).

It should be noted that the daytime values of the $I_H$ index in the Chamois village, even considering the complete absence of cars, always remain at around 5 (even in the low autumn season), due to human maintenance activities of the territory and the contribution, to the background sound level, of a stream that flows constantly throughout the year in this area.

Contrary to what could be expected for the Cheneil tourist resort, which offers a timeless natural environment, characterized by the unspoilt countryside of this valley, with dense fragrant woodlands, alpine pastures, and no ski infrastructures, the event component (EVT) has been found to be high for the entire season, due to the repeated barking of dogs in the area.

The results of the evaluation of the $I_H$ index for the nighttime period are shown in Table 14.

**Table 14.** The nighttime $I_H$ index (22-06 h) for the three seasons for all the monitored resorts.

| Site | Night Period (22-06) | | | | | | | | |
| | Summer | | | Autumn | | | Winter | | |
| | bgn | evt | $I_H$ | bgn | evt | $I_H$ | bgn | evt | $I_H$ |
|---|---|---|---|---|---|---|---|---|---|
| Chamois village | 3.7 | 0.4 | 4.1 | 3.5 | 0.4 | 3.9 | 2.9 | 0.5 | 3.4 |
| Chamois-Lake Lod in the ski area | 3.4 | 0.7 | 4.1 | 0.0 | 0.6 | 0.6 | 0.0 | 1.1 | 1.1 |
| Chamois-Teppa, high ski area | 0.0 | 1.7 | 1.7 | 0.0 | 1.8 | 1.8 | 0.0 | 1.1 | 1.1 |
| Cervinia-church square | 4.0 | 0.3 | 4.4 | 3.6 | 0.4 | 4.0 | 0.8 | 4.9 | 5.7 |
| Cervinia-Giomein place | 4.3 | 0.3 | 4.6 | 3.5 | 0.4 | 3.9 | 1.6 | 1.5 | 3.1 |
| Cervinia ringroad | 2.7 | 0.9 | 3.6 | 2.1 | 0.5 | 2.6 | 0.0 | 4.3 | 4.3 |
| Plan Maison-middle station ski area | 1.4 | 0.3 | 1.8 | 0.2 | 0.3 | 0.5 | 0.0 | 5.6 | 5.6 |
| CimeBianche-high ski area | 0.6 | 0.6 | 1.2 | 0.0 | 1.0 | 1.0 | 1.1 | 3.7 | 4.8 |
| Plateau Rosà station on the glacier | 0.9 | 0.5 | 1.4 | 0.0 | 8.3 | 8.3 | 1.5 | 2.3 | 3.8 |
| Cheneil-rural village and pastures | 1.8 | 3.0 | 4.8 | 0.7 | 3 | 3.7 | / | / | / |

There is generally a good natural acoustic climate in the sites outside built-up areas constituted by mountain pastures and woods, such as in the ski areas that are not affected by individual natural sound sources from streams or by noise from the bottom of the valley (Teppa, Plan Maison, Cime Bianche, Plateau Rosà) in all seasons and in particular during the night (the background levels are very low and at the limit of instrumental detection).

There is also an absence of particularly noisy events in these areas, except during winter, when slope maintenance operations can take place in some sites.

However, it should be noted that most of these maintenance operations take place in areas with few dwellings, apart from a few chalets or hotels at a high altitude, and can be programmed in order to limit the disturbance.

The values of the $I_H$ index receive a greater contribution from background sound (BGN) in summer and, in some cases, also in early autumn, due to the various natural sources (chirping birds, roaring water) and to a lower and quieter attendance of tourists in these mountain areas than in the winter season.

The only case where a value was found to exceed the critical threshold occurred, paradoxically, in the most remote site, which is located at 3500 m (a.s.l.) on the Plateau Rosà glacier. However, this was due to events produced by the wind, which blew at speeds of about 5 m/s around the ski infrastructures in the area.

Several studies carried out over the last decade have shown that a multidisciplinary approach is more appropriate for the analysis of the perception of sound than focusing only on the sound pressure level [27]. For this reason, after the acoustic analysis, another investigation was conducted at the Plan Maison location: the evaluation of the psychoacoustical parameters [28,29] on the basis of the audio files acquired during the short-term measurements. This area is located in a natural environment, but it is influenced by human mountain activities in some seasons. The sound samples used for the evaluation include the entire sound recordings of the short-term measurements, without any diversification between natural environmental sounds (water, wind, birdsongs) and artificial ones (music, skilifts, snow makers). The psychoacoustical parameters fluctuation strength, loudness and sharpness were calculated for each of the seasonal short-term measurements. It has in fact been widely reported that loudness and sharpness are closely correlated with soundscape perceptions, and they could be appropriate indicators to discriminate between sound source types in mountain areas.

However, it should be noted that the main objective of the study was not the analysis of the psychoacoustical parameters. Furthermore, the results of the first elaborations were not sufficiently significant and did not provide any additional information on the seasonal variations of the acoustic climate in the area: they have therefore not been included in this paper and it will be necessary to explore different sound categories to obtain meaningful information from them [30]. These aspects will certainly be analyzed in future studies in order to integrate the current methodology.

## 5. Conclusions

Since univocal criteria for the acoustical characterization of open country areas are not yet available, the results of all the indicators quantified by ARPA-VdA over the years, in different types of mountain resorts, will be compared in the future in order to identify the best methodology to apply in a mountain context.

The studies conducted in Europe until now have taken into account both acoustic parameters and the related human perception of noise as a function of the considered area. In addition, geographical criteria, based on the distance from the main sound sources and cover land use, have recently been used for the quantification of the Quietness Suitability Index (QSI) [31].

The methodology applied in this study may be extended to other similar touristic resorts and to other open countryside areas, which are affected more by noise from the main infrastructures present on the bottom of the valleys. The present results, which are based on acoustic field measurements, can be coupled with other geographical indices, such as the QSI, to eventually arrive at the delimitation of quiet mountains areas, considering the sources of noise related to tourism activities.

As in the case of the Harmonica Project (http://www.noiseineu.eu), interviews with mountain users may be conducted to obtain feedback on the comprehensibility of the $I_H$ index. At the same time, the psychoacoustic analyses will be dealt with in more detail with reference to particular sources from

the sports and tourism activities (heli-skiing, mountain biking, mountain climbing, trekking) practised in these mountain areas.

The acoustic measurements and analyses carried out over the years and in this study by ARPA-VdA confirm the need for accurate and specific evaluations for the characterization of highly natural mountain areas, with particular reference to seasonal variations related to the consequent different tourist attendances [32].

The measurement methodology and data processing were effective, and in general, the positions of the microphones were suitable for the seasonal acoustic characterization of the various sites. The encountered difficulties were mainly related to the problems involved in reaching the most remote sites in autumn when the ski lifts are closed or in periods when heavy snowfall and low temperatures make it impossible to reach the sites.

As far as the latter point is concerned, it will be interesting to implement the collection of sound levels, using low-cost digital technology, mobile applications, and data made available by users, in order to extend the studied territory and to simplify the monitoring methodology [33].

The correlation between the fixed station sound levels and those from the relative short-term measurement stations was found to be good for the background levels quantified by LAeq95, but less significant for LAeq, due to individual events that take place in these mountain sites.

The results of the acoustic measurements confirmed all the results that were expected when the study was defined, and in particular:

- The three tourist resorts of Chamois, Cheneil and Breuil-Cervinia, although having similar landscape features, have a different soundscape, depending on the period of year and on the presence of tourists.
- Seasonal acoustic climate alterations were evident in all the studied areas and are due to the contribution of both natural sources (streams, insects, animals, etc.) and to noisy human events that do not normally take place in a natural mountain environment, but which are linked to the skiing activity of people from all over the world. Music in bars or pubs, ski lift operations, the maintenance of ski slopes, artificial snowmaking, helicopters to deliver supplies to alpine shelters or for heli-skiing transport, are typical examples of such noise sources.

The Cheneil valley is unlike the other two resorts as far as these two aspects are concerned, because the mountain and pastures in this location have been safeguarded against mass tourism, especially regarding the absence of noisy, artificial infrastructures.

Noise from work equipment and the machinery used for construction activities in the village and for the ski slope maintenance (bulldozers, hammers, etc.) has to be added to the artificial sound sources described above in the autumn.

In general, the most common natural acoustic sources are from water, birds, and insects in summer, streams in autumn, and the wind in winter, which is the quietest season because the presence of snow dampens almost all the natural sounds. In mountain sites, and in particular in ski resorts located at higher altitudes (above 1800 m a.s.l.), winter blends with spring, and snow on the ground lasts until just before the beginning of summer.

$I_H$, a noise pollution index developed to provide information that is easier to read and understand by people in general, allows a quick and easy seasonal or day–night comparison of the sound levels in mountain areas where ski resorts are located. The $I_H$ index also represents a specific elaboration of the basic acoustic parameters which could allow comparisons to be made with other monitored sites in parks and green areas in different European regions and cities.

Moreover, taking into account both background sound levels and the related levels of noisy events, the index makes it possible to quickly understand how much humans can alter the natural acoustic climate of an area in order to create their entertainment.

As already mentioned in Section 2.7, the formula of the $I_H$ index, which was calibrated in the HARMONICA project for background sound levels above 30 dBA, has not been modified in the present

study. As suggested by the index developers, in order to remain in line with the philosophy of the $I_H$ index, when the calculation gave negative values (background sound levels below 30 dBA, which are typical of winter), the BGN component of the index, related to the background level, was set to zero. Therefore, in such cases, the index was calculated taking into account only the EVT contribution (acoustic energy provided by sound peaks that emerge above the background level).

The main logic behind the index definition was the absence of any noise-related health effect in any context for values below 30 dBA, and on the common-sense concept that a constant sound level of 30 dBA could represent an ideal level below which a detailed quantification is not necessary. This aspect certainly deserves further investigation in order to adapt the index to remote mountain areas in a more realistic manner.

The results of all the elaborations and evaluations carried out in each measurement site are reported in the text of the complete study at the following address https://webthesis.biblio.polito.it/7856/.

**Author Contributions:** Conceptualization, C.T. (Christian Tibone), G.T., M.M., and M.C.B.; data curation, C.T. (Christian Tibone), G.T., D.C., and C.T. (Christian Tartin); formal analysis, G.T., M.M., and C.T. (Christian Tibone); investigation, C.T. (Christian Tibone), G.T., D.C., C.T. (Christian Tartin), and F.B.; methodology, G.T., M.M., C.T. (Christian Tibone), and M.C.B.; validation, M.M., G.T., and C.T. (Christian Tibone); supervision, M.M., M.C.B., and G.A.; writing—original draft preparation, C.T. (Christian Tibone), G.T., M.M., and F.B.; writing—review and editing M.M., C.T. (Christian Tibone), F.B., and G.T. All authors have read and agreed to the published version of the manuscript.

**Funding:** This research received no external funding.

**Acknowledgments:** The authors wish to thank Louena Shtrepi and Francesco Aletta for the stimulating discussions on the psychoacoustic indexes. The authors would also like to thank Cervino S.p.a. and the Valtournenche and Chamois Municipality Authorities for allowing access to the acoustic monitoring sites.

**Conflicts of Interest:** The authors declare no conflicts of interest.

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
