# Peer review of "Seasonal Variability of the Acoustic Climate of Ski Resorts in the Aosta Valley Territory"

_environments, doi:10.3390/environments7030018_

Round 1

Reviewer 1 Report

The contents of the text is a well presented engineering report aimed at sound measurement and analysis in three locations of the Italian Alps. The authors clearly aim at "to characterize and analyze the sound climate of the urban centers" but seem to fail to highlight or incorporate innovative aspects that may lead to their contribution to the state of the art.

The authors mix up the concepts of sound and noise as in lines 55-65. Here they mention "natural environmental noise levels" which is an awkward expression. Certainly they mean sound pressure levels. And if they refer to natural sources they should mean sound and not noise, or are there negative connotations or perception (associated nuisance?). Perception aspects seem actually to be absent (except for the last paragraph of the Discussion section), but there are a number of references to soundscape in the text. There are also no references to sound/noise exposure of residents.

Author Response

Point 1:The contents of the text is a well presented engineering report aimed at sound measurement and analysis in three locations of the Italian Alps. The authors clearly aim at "to characterize and analyze the sound climate of the urban centers" but seem to fail to highlight or incorporate innovative aspects that may lead to their contribution to the state of the art.

Response 1:The introduction section has been expanded, highlighting the goals and innovative aspects of the paper.

Point 2:The authors mix up the concepts of sound and noise as in lines 55-65. Here they mention "natural environmental noise levels" which is an awkward expression. Certainly they mean sound pressure levels. And if they refer to natural sources they should mean sound and not noise, or are there negative connotations or perception (associated nuisance?).

Response 2: The wording has been thoroughly revised throughout the paper, restricting the use of the word “noise” to pertinent cases (nuisance related).

Point 3: Perception aspects seem actually to be absent (except for the last paragraph of the Discussion section), but there are a number of references to soundscape in the text.

Response 3: We have removed 3 documents on soundscape in the references (number 5, 10 and 25).

Point 4:There are also no references to sound/noise exposure of residents.

Response 4:The focus of the paper is on the acoustic climate of natural / touristic areas. Therefore, it provides no explicit reference  to exposure of residents. This is also because the investigated sound sources (ski-related equipment, agricultural / pasture related equipment, etc.) have usually no significant impact on dwellings.

Reviewer 2 Report

I accept the manuscript without having to make corrections by the author.

The impacts of artificial sources on the environmental noise need to be considered case by case, not referring to a fixed standard, but assuming the natural environmental noise in a given site as the standard for the site itself.

This study contributes knowledge on the characterization of the seasonal variation of the acoustic climate of three tourist resorts in Alps. It is very interesting. The study aims at developing a method for the acoustic characterization of ski resorts through the identification of suitable indicators to be applied to the noise levels detected.

In addition to the collection of acoustic data in remote areas, the study is intended to provide additional criteria for the selection and management of quiet areas in open space.

I think the article is easy to read.

Author Response

Point 1:The impacts of artificial sources on the environmental noise need to be considered case by case, not referring to a fixed standard, but assuming the natural environmental noise in a given site as the standard for the site itself.

Response 1: Wording has been revised.

Point 2: This study contributes knowledge on the characterization of the seasonal variation of the acoustic climate of three tourist resorts in Alps. It is very interesting. The study aims at developing a method for the acoustic characterization of ski resorts through the identification of suitable indicators to be applied to the noise levels detected.

Response 2: OK.

Point 3: In addition to the collection of acoustic data in remote areas, the study is intended to provide additional criteria for the selection and management of quiet areas in open space.

Response 3: OK.

Reviewer 3 Report

The determination of soundscape in every kind of rest areas is very important. This article describes an acoustic climate in chosen ski areas. The research value of work is developed. The goal should be properly formulated and the scientific aspect, the novelty, should be emphasized.

The Introduction makes references to some acoustic, soundscape studies, as well as a few noise studies. Please, change and extend the Introduction (e.g. works of Antonella Radicchi or Kephalopoulos, Stylianos et al. "Advances in the development of common noise assessment methods in Europe: The CNOSSOS-EU framework for strategic environmental noise mapping." or Cassina L. et al. (2018). Audio-visual preferences and tranquillity ratings in urban areas.). The reviewer suggests moving the beginning of the Introduction (about research areas - lines 35-51) to Materials and Method.  Please, correct the quality of figures 1-9 and 11-13 (e.g. Figure 1 is illegible - correct the map, I know - exists web page but if You want to present research areas - do it better). The Figures require sources (Google Maps? other?). Please, to mark clearly the article's goal - in the Abstract, Introduction. What is the novelty of the article? What are the innovations? Please, to emphasize the importance of IH. Please, check English - spelling, and comas.

Author Response

Point 1: The determination of soundscape in every kind of rest areas is very important. This article describes an acoustic climate in chosen ski areas. The research value of work is developed. The goal should be properly formulated and the scientific aspect, the novelty, should be emphasized.

Response 1: The introduction section has been expanded, highlighting the goals and innovative aspects of the paper.

Point 2: The Introduction makes references to some acoustic, soundscape studies, as well as a few noise studies. Please, change and extend the Introduction (e.g. works of Antonella Radicchi or Kephalopoulos, Stylianos et al. "Advances in the development of common noise assessment methods in Europe: The CNOSSOS-EU framework for strategic environmental noise mapping." or Cassina L. et al. (2018). Audio-visual preferences and tranquillity ratings in urban areas.).

Response 2: Suggestion accepted adding 2 works. The first of Kephalopoulos at al. (reference number 3) and the second of Antonella Radicchi (reference number 33).

Point 3: The reviewer suggests moving the beginning of the Introduction (about research areas - lines 35-51) to Materials and Method.

Response 3: Suggestion accepted.

Point 4: Please, correct the quality of figures 1-9 and 11-13 (e.g. Figure 1 is illegible - correct the map, I know - exists web page but if You want to present research areas - do it better).

Response 4: All the mentioned figures have been revised or redrawn for a better legibility.

Point 5: The Figures require sources (Google Maps? other?).

Response 5: Sources of the maps have been indicated as footnotes in the captions.

Point 6: Please, to mark clearly the article's goal - in the Abstract, Introduction. What is the novelty of the article? What are the innovations? Please, to emphasize the importance of IH.

Response 6: The abstract and the introduction have been revised and expanded.

Round 2

Reviewer 1 Report

The authors have adequately addressed most previous comments. It is still not obvious what innovation this paper brings. However, the article is well written and this version made clearer to the reader the contributions of the contents to knowledge.

The authors should still clarify some inconsistencies in the text

on lines 280/281 the authors write "the environment has a good soundscape". How do you qualify "good"? Since the "soundscape" is understood, according to ISO 129213-1:2013, as the acoustic environment as perceived and the authors report that perception issues were not contemplated, that statement should be clarified; on lines 382/383 where the authors write "the equivalent levels" they probably mean "the continuous equivalent sound pressure levels", or the LAeq levels; on lines 497/498 on the sentence "the snow on the ground covers almost the entire sound" it is not clear what the authors mean by "covers" - if they refer to sound absorption, this should be made clear.

Some minor typos and English grammar issues should also be corrected.

Author Response

Point 1:The authors have adequately addressed most previous comments. It is still not obvious what innovation this paper brings. However, the article is well written and this version made clearer to the reader the contributions of the contents to knowledge.

Response 1: OK.

Point 2: The authors should still clarify some inconsistencies in the text.

Response 2: The inconsistencies have been clarified simplifying the text.

Point 3:on lines 280/281 the authors write "the environment has a good soundscape". How do you qualify "good"? Since the "soundscape" is understood, according to ISO 129213-1:2013, as the acoustic environment as perceived and the authors report that perception issues were not contemplated, that statement should be clarified; on lines 382/383 where the authors write "the equivalent levels" they probably mean "the continuous equivalent sound pressure levels", or the LAeq levels; on lines 497/498 on the sentence "the snow on the ground covers almost the entire sound" it is not clear what the authors mean by "covers" - if they refer to sound absorption, this should be made clear.

Response 3: In the new revision sent: On lines 285/286 the word soundscape has been changed in acoustic climate more relevant to sound levels;

On lines 386/387 the expression ” the equivalent levels” has been changed with “LAeq levels”

On line 531 on the sentence “the snow on the ground covers almost ... “we have added “all the natural sources sound”. There is no reference to sound absorption.

Point 4: Some minor typos and English grammar issues should also be corrected

Response 4: Typos and English grammar issues have been corrected from professional English speaking editor.

Reviewer 3 Report

Dear Authors,

Thank you for your reference to the reviewer's comments and corrections made. In the following articles, please pay attention to the length and transparency of the sentences.
Good luck.

Author Response

Point 1:Dear Authors,Thank you for your reference to the reviewer's comments and corrections made. In the following articles, please pay attention to the length and transparency of the sentences.Good luck.

Response 1: We have checked the test and we have changed the length of  any sentences.